# A Large-Scale Multiple Genome Comparison of Acidophilic Archaea (pH ≤ 5.0) Extends Our Understanding of Oxidative Stress Responses in Polyextreme Environments

**DOI:** 10.3390/antiox11010059

**Published:** 2021-12-28

**Authors:** Gonzalo Neira, Eva Vergara, Diego Cortez, David S. Holmes

**Affiliations:** 1Center for Bioinformatics and Genome Biology, Fundación Ciencia & Vida, Santiago 7780272, Chile; gonzalo.neira@ug.uchile.cl (G.N.); evergara@cienciavida.org (E.V.); dnahuel@cienciavida.org (D.C.); 2Facultad de Medicina y Ciencias, Universidad San Sebastián, Santiago 8420524, Chile

**Keywords:** comparative genomics, catalase, peroxiredoxin, superoxide dismutase (SOD), superoxide reductase (SOR), rubrerythrin, antioxidant enzymes, oxidative stress, reactive oxygen species (ROS), horizontal hene transfer (HGT)

## Abstract

Acidophilic archaea thrive in anaerobic and aerobic low pH environments (pH < 5) rich in dissolved heavy metals that exacerbate stress caused by the production of reactive oxygen species (ROS) such as hydrogen peroxide (H_2_O_2_), hydroxyl radical (·OH) and superoxide (O_2_^−^). ROS react with lipids, proteins and nucleic acids causing oxidative stress and damage that can lead to cell death. Herein, genes and mechanisms potentially involved in ROS mitigation are predicted in over 200 genomes of acidophilic archaea with sequenced genomes. These organisms are often be subjected to simultaneous multiple stresses such as high temperature, high salinity, low pH and high heavy metal loads. Some of the topics addressed include: (1) the phylogenomic distribution of these genes and what this can tell us about the evolution of these mechanisms in acidophilic archaea; (2) key differences in genes and mechanisms used by acidophilic versus non-acidophilic archaea and between acidophilic archaea and acidophilic bacteria and (3) how comparative genomic analysis predicts novel genes or pathways involved in oxidative stress responses in archaea and likely horizontal gene transfer (HGT) events.

## 1. Introduction

Reactive oxygen species (ROS) are formed as byproducts of aerobic metabolism including, for example, Fenton chemistry [1,2] and can involve molecules such as the superoxide anion (O_2_^–^), hydrogen peroxide (H_2_O_2_) and the hydroxyl radical (·OH). The accumulation of these molecules can lead to oxidative stress and produce damage to many cellular macromolecules, such as DNA and RNA, as well as to proteins that can impair the function of important cellular structures such as membranes [3,4,5,6]. To mitigate this damage, different mechanisms have been developed by prokaryotes. These include superoxide scavengers such as superoxide dismutase (SOD) and superoxide reductase (SOR), which can diminish the quantity of superoxide anion [7]. Peroxiredoxins (Prx) and catalases are examples of peroxide scavengers that reduce the levels of hydrogen peroxide [8].

Whereas extensive research has been carried out on oxidative stress mechanisms in the bacterial domain [9,10], including the report of an inventory of genes across the whole domain [11], less is known about stress responses in archaea. We aim to identify the potential mechanisms of oxidative stress response in extremely acidophilic archaea and to analyze the phylogenetic distribution of predicted response mechanisms across the domain.

Since ancestral archaea have been proposed to have emerged in early Earth environments [12,13], their study may provide useful insight into the evolution of oxidative stress responses. Early evolution most likely proceeded in anaerobic conditions [14], but archaea may have been exposed to “whiffs” of oxygen that led to the need for first oxidative stress response mechanisms [15,16], then additional mechanisms may have evolved as they transitioned from the anaerobic world to an aerobic atmosphere during the great oxidation event (GOE) 2.4 billion year ago [17].

The archaeal domain comprises a wide range of organisms, many of which thrive in environments that are considered polyextreme, such as extremely high temperatures, low and high pHs and high salinity conditions [18,19,20]. Within these groups, stress response mechanisms have been described for hyperthermophiles [21], which constitute the majority of all identified archaeal organisms. Acidophiles are commonly defined as organisms that grow optimally at pH lower than 5 [22,23]. Moderate acidophiles are defined as organisms that grow optimally between pH 3 and pH 5 [24], and extreme acidophiles are organisms that grow optimally at pH values lower than 3 [25]. The environmental constraints each group faces are particular to their pH range [26].

One particularly challenging constraint that acidophiles face is the unusually high concentration of dissolved metals and metalloids, such as Pb, Zn, Cu, Ni, Cr, Cd, Mn, Mo and Fe [27,28,29,30], in their environments, where the concentration of bioavailable Fe(II) in econiches at pH 3 can be 18 orders of magnitude greater than that encountered in circumneutral environments [31].

Previous studies of oxidative stress responses of acidophilic archaea demonstrated the presence of several mechanisms that have also been reported for other archaea, such as superoxide dismutase, peroxiredoxin and rubrerythrin [32,33,34]. Interestingly, a specific alkyl hydroperoxide reductase has been described in some thermoacidophilic archaea, such as the strict aerobic, hyperacidophilic (pH < 1) *Picrophilus torridus* (phylum Euryarchaeota) that has not been detected in non-acidophilic archaea [35].

However, despite this prior research, a large scale, more comprehensive understanding of stress-associated mechanisms in acidophilic archaea has not been undertaken. Our report begins to fill in this lacuna in our knowledge, providing a more extensive inventory of oxidative stress responses and helping to identify mechanisms provoked by multiple simultaneous stresses such as high temperature, extreme acidity and high metal loads. Since archaea have been proposed to root at the base of the tree of life, their study may provide useful insight into the evolution of oxidative stress responses on exposure to regular oxygen intrusion as the world transitioned from an anaerobic to aerobic atmosphere.

## 2. Materials and Methods

### 2.1. Genome Collection

A dataset comprising 234 acidophilic (optimal growth pH < 5) archaeal genomes was obtained from the database AciDB [36]. All genomes were downloaded from the National Center for Biotechnology Information (NCBI) (https://www.ncbi.nlm.nih.gov/genome/) and the Joint Genome Institute (JGI) (https://img.jgi.doe.gov/) both accessed on the 30 of October 2020. Only high-quality genomes (draft or complete) were considered for the analyses, defined as a >80% completeness and <5% contamination as calculated by CheckM v1.0.12 with the standard lineage workflow [37]. Existing genome annotations in Refseq [38] were used when available and, for genomes with no annotation, Prokka v1.13.3 was used to identify genomic features [39]. Growth condition metadata for each genome was obtained from AciDB using the optimum values for pH and temperature. Oxygen requirement information was collected from bibliographic research associated with each species. If the data for a specific species were not available, they was extrapolated from the genus description. This led to a classification of all organisms into aerobic, anaerobic or facultative. A summary of genomic information is provided in Appendix A.

### 2.2. Phylogeny

Phylogenetic reconstruction was performed with PhylophlAn 3.0 using the provided dataset of 400 conserved proteins [40]. The parameters were set as the following: diversity as high (as recommended for phylum wide phylogenies); marker search performed using diamond (v2.0.2) [41]; sequence alignment by MAFFT (v7.310) with the L-INSI iterative refinement option [42]; and alignment trimming by trimAl (v1.2) [43]. Tree construction was performed using IQTREE v1.6.1. The best evolutionary model was identified using ModelFinder according to the Bayesian information criterion (BIC) and Akaike information criterion (AIC) [44]. To assess the support of each branch, the non-parametric ultrafast bootstrap method (with 1000 replicates) was used [45]. Phylogenetic trees were visualized and annotated using iTOL [46]. A species clustering analysis was performed using FastANI with a value of 95% identity to classify genomes as the same or different species [47].

### 2.3. Functional Annotation and Orthologous Groups Identification

A set of 52 proteins associated with oxidative stress responses was identified from the literature. Functional annotations based on Pfam [48] and InterPro families (domain and superfamily) [49] for each protein were obtained (Appendix A). All genomes in the study were functionally annotated using the InterProScan pipeline with default parameters [50]. Orthogroups defined as proteins that have evolved from a common ancestor and include both orthologs (homologous protein from two species) and paralogs (a set of proteins that have a common origin in the same genome) were identified using OrthoFinder v2.3.3 [51]. Protein sequences were compared using DIAMOND in an all-versus-all search and orthogroups were inferred [41]. DendroBLAST [52] was used to generate unrooted gene trees for each of the identified orthogroups. MAFFT was used for multiple sequence alignments and FastTree [53] was used for the tree inference. All other steps were performed with default parameters. An in-house Python script was used to identify oxidative stress response proteins in the acidophilic genomes based on the functional annotation. Pfam annotations provided a broad scope view of candidates, which were subsequently hand-curated by InterPro family and domain annotation to obtain additional information. Subcellular locations were assigned to predicted proteins and classified as cytoplasmatic, inner membrane, exported, outer membrane or periplasmic using PSORTb v3.0 [54], and signal peptide identification was performed using SignalP v5.0 [55].

### 2.4. Evolutionary Trajectory Analysis

An evolutionary history-based (phylogenetic) approach as described by Ravenhall et al. [56] was used to identify genes whose evolutionary history significantly differs from that of the host species and are inferred to be horizontally transferred genes (HGT). Phylogenetic trees were constructed using the sequences in the orthogroups and their best blastp hits as described by Nelson et al. [57] and Kooning et al. [58]. Proteins from the blastp search were downloaded using the Batch Entrez web tool from NCBI [59]. Sequence alignments were performed using MAFFT and visualized with AliView v1.2.6 [60] to identify the conservation of key amino acids when needed. The phylogenetic tree was then constructed using IQTREE with an ultrafast bootstrap of 1000 replicates. Phylogenetic trees were visualized and clades were annotated using iTOL [46]. Genome neighborhoods were analyzed using Gene Graphics with a region size of 5000 bp [61].

## 3. Results

### 3.1. Phylogeny and Species Clustering

High-quality draft and complete genomes from acidophilic archaea were obtained from AciDB [36] with a total of 180 different genomes that had over 80% completeness and less than 5% contamination as defined by CheckM [37] Of these genomes, 47 were complete and 133 were permanent drafts (Appendix A). A phylogenetic tree was inferred using a set of 400 conserved markers as explained in Materials and Methods. Three organisms from the ARMAN clade (Ca. *Mancarchaeum acidiphilum* Mia14, *Thermoplasmatales archaeon* A_DKE and Ca. *Micrarchaeum* sp. AZ1) were excluded from the tree inference as these organisms lack the majority of the conserved markers. Four different phyla are represented in the phylogenetic tree: Crenarchaeota, Euryarchaeota, Marsarchaeota and Thaumarchaeota (Figure 1). At the genus level, Crenarchaeota and Euryarchaeota were the most diverse phyla in the dataset with 10 and 6 different genera, respectively. The species clustering analysis based on ANI values reveals that *Cuniculiplasma divulgatum* is the Euryarchaeota with the most genomes associated (19 in total), as a series of genomes annotated as “*Thermoplasmatales archaeon*” were identified as part of this species. *Sulfolobus acidocaldarius* from Crenarchaeota has the most genomes of any species with 54 genomes associated with it. These results are summarized in Appendix A. The lack of markers in the ARMAN clade organisms is not surprising as this clade is characterized by their small genome sizes derived from their symbiotic lifestyle [62]. The species clustering highlights the importance of developing new standards in the naming of prokaryotic genomes [63,64] that are not from isolated sources, as the adoption of higher rank taxonomic names may lead to confusion in subsequent analysis.

### 3.2. Superoxide Defense Proteins

Superoxide anion (O_2_^−^) is a highly reactive species that can cause severe damage inside the cell. The first system used in the process of removing O_2_ is superoxide dismutase (SOD), which produces hydrogen peroxide and oxygen. Superoxide dismutase is classified into four different categories depending on the cofactor used (Mn, Fe, Cu/Zn and Ni). The second system is associated with the superoxide reductase (SOR), which produces hydrogen peroxide and protons in the reduction of superoxide. Details of how both mechanisms work are reviewed by Sheng and colleagues [65].

#### 3.2.1. Superoxide Dismutase (SOD)

We found a wide distribution of Fe-SOD throughout several acidophilic archaeal genera from all phyla under study, despite it previously being identified in only a few organisms, such as *Saccharolobus solfataricus* [66], *Ferroplasma acidiphilum* [67] and *Metallosphaera sedula* [68] (Appendix A). The abundance of this category of SOD (Fe-SOD type) may be of evolutionary importance given that the iron-containing isoform has been suggested to be an ancient type of SOD [7]. The presence of Fe-SOD in acidophiles is interesting as Mn-SOD or cambialistic SODS (that use either Fe or Mn) have been previously reported in several other archaea [69,70]. The higher availability of iron in low pH environments, together with the notion that archaea are ancestral organisms [12], highlight these acidophiles as prime candidates to be the original source of one of the most relevant oxidative stress mechanisms used across all domains of life for evolutionary studies and biotechnological applications. Phylogenetic analysis of the predicted sequences was performed to study the evolution of SOD in these organisms. The sequences were divided into four different clades: two clades that grouped most of the sequences associated with the phylum Crenarchaeota and Euryarchaeota and two smaller clades formed by sequences of organisms from the phylum Marsarchaeota and Thaumarchaeota, the latter including outliers of the Crenarchaeota phylum from the *Acidilobus* and *Caldisphaera* genera. The distribution in each of the separated clades appears to be associated with phylogenetic relatedness, as each predicted protein branch is formed mainly by organisms from the same genus. There is a clear distinction between the sequences from Euryarchaeota and those from the phyla Crenarchaeota, Thaumarchaeota and Marsarchaeota, which are phylogenetically part of the TACK superphylum [71] (Figure 2).

An exception to the phylogenetic association can be observed for Ca. *Micrarchaeum* sp. AZ1 and Ca. *Mancarchaeum acidiphilum* Mia14 sequences, which are both inside the Crenarchaeota clade, with the former close to *Thermoproteus* and the latter to *Acidianus* (Figure 2, red asterisks). This difference could be explained by the fact that Micrarchaeota organisms have reduced genome size and limited metabolic capabilities, relying on other members of the community to supply these deficiencies by a symbiotic approach [72]. The observation that all SODs from these organisms are closely related with organisms that thrive in the same environments suggest that this oxidative stress mechanism was likely gained by HGT, a process that is essential in the evolution of other members of the DPANN clade [73].

#### 3.2.2. Superoxide Reductase (SOR)

The superoxide reductase (SOR) was predicted only in the genus *Aciduliprofundum*, which is the only genus that lacks any SOD sequence. Phylogenetic analysis of the sequences identified in *Aciduliprofundum* and archaeal proteins from the Superoxide Reductase Gene Ontology Database (SORGOdb) [74] shows a clade formed with other strict anaerobes from Euryarchaeota (non-acidophiles) falling under the class II related SOR. This suggests that vertical transmission could play a key role in the evolution of this protein (Appendix A). Previous studies have shown that SOR is mainly found in anaerobic organisms [75], such as *Aciduliprofundum*, which is of interest in relation to other anaerobic organisms in the study (such as *Vulcanisaeta* and *Acidilobus*) that use Fe-SOD instead as the main medium to remove O_2_. Rubredoxin (Rb) has been identified as an electron donor for the reduction of superoxide via SOR in *Pyrococcus furiosus* (also from the Euryarchaeota phylum) [76]. It was also identified only in *Aciduliprofundum*, supporting the idea that this organism is the only one in the dataset able to reduce superoxide. One possible explanation for this contradiction with previous studies that claim that the main mechanism for coping with superoxide in anaerobic organisms is the use of SOR (avoiding the production of O_2_ resulting from SOD), during which acidophiles thrive. SOR in the reduction of O_2_ produces extra protons (that are not produced by SOD), leading to additional stress for acidophilic organisms that need to maintain a neutral intracellular pH and balance the extremely high concentration of protons in the environment. This is a disadvantage of the use of SOR in low pH environments that favor the use of SOD. SOD has previously been observed as one of the main oxidative stress response mechanisms in acidophiles [77] and results in the loss of the SOR gene through adaptation to the harsh environmental conditions.

### 3.3. Peroxide Scavengers

#### 3.3.1. Catalase

Catalases convert hydrogen peroxide into water and oxygen and work as a common oxidative stress mechanism in organisms from all domains of life [11]. In acidophilic archaea, only genomes from Thermoplasmatales and *Aciduliprofundum* sp. MAR08-339 contain predicted catalase-peroxidase (with the same functional annotation as KatG). Phylogenetic analyses with proteins from NCBI show that the sequences from Thermoplasmatales form a clade close to extremely acidophilic bacteria from the genus *Methylacidimicrobium*, which are also moderate thermophilic organisms. KatG identified for *Aciduliprofundum* sp. MAR08-33 forms a separated clade with other Euryarchaeota organisms (not acidophiles) and some Deltaproteobacteria (Appendix A). Their low presence in archaea has been reported previously, as organisms that have SOD usually do not have catalases [21], and, as previously discussed, we identified SOD in most acidophilic genomes, with the exception of *Aciduliprofundum*. The result of mixed origin clades in the phylogenetic reconstruction where archaea and bacteria share the same clade suggests that HGT between domains may play a major role in the evolution of catalases. This aligns with studies that show that the phylogeny of catalases is complex and often does not follow a pattern of vertical inheritance [78,79].

#### 3.3.2. Osmotically Inducible Protein C (OsmC)

Another protein with peroxide scavenging activity is the osmotically inducible protein C (OsmC), which is involved in the elimination of organic peroxides in bacteria [80]. OsmC was found to be ubiquitous in acidophilic archaea, with most organisms in Crenarchaeota, Euryarchaeota and Marsarchaeota predicted to have one or more copies, even reaching three copies in *Metallosphaera* organisms. Phylogenetic tree analysis of the identified sequences reveals a clear division between the sequences that evolved from Euryarchaeota and Crenarchaeota. Nine orthologous groups were formed by the predicted OsmC sequences. The orthogroups mostly match the clade as observed in Appendix A, where each orthogroup contains proteins from the same phylum except for OG2461 and OG3045, which also contain sequences from Marsarchaeota. Protein sequences from Marsarchaeota were distributed across clades with sequences from both Crenarchaeota and Euryarchaeota, forming separated clades, suggesting HGT between these phyla. KAD1 in *Thermococcus kodakarensis* is an OsmC protein that can reduce both inorganic and organic peroxides [81]. It is interesting to note that the different orthogroups identified across acidophilic archaea may be related with functional distinction to the classic OsmC. Further experimental investigation is needed to determine unravel the functional activities of these different identified predicted proteins.

#### 3.3.3. DNA-Binding Protein from Starved Cells

Another way to prevent the creation of ROS is sequestering iron, thus decreasing the Fenton reactions. A protein that fulfills this function is the DNA-binding protein from starved cells (Dps), a member of the ferritin-like superfamily that can also reduce hydrogen peroxide in the process of oxidation of free ferrous iron [82]. Dps in archaea can be divided into two different functional annotations, the first with the InterPro family IPR002177 group from the model protein for Dps in bacteria and the second with the InterPro family IPR014490 domain that represents Dps-like proteins from archaea first identified in *Saccharolobus solfataricus* [83]. Using the InterPro family IPR002177 domain as a probe (bacteria associated), we predicted Dps in Marsarchaeota and *Sulfolobales* MK5 as part of a single orthogroup. Unlike the previous case, the Dps-like family (archaea associated) was identified in three different phyla: Crenarchaeota (*Metallosphaera*, *Saccharolobus* and *Sulfolobus*), Euryarchaeota (*Aciduliprofundum*) and Thaumarchaeota (Ca. *Nitrosotalea*). To study the possible evolutionary origin of the Dps protein, a phylogenetic tree was constructed using the best hits from a blastp search. The Dps phylogeny shows their closest sequences include other acidophiles coming from Acidobacteria and several Proteobacteria, suggesting cross-domain HGT as the main evolutionary force for this protein in archaea that may be associated with a specific econiche adaptation [84] (Figure 3).

In contrast, Dps-like protein is widely found in archaea from Crenarchaeota and Thaumarchaeota. The functionality of the Dps-like protein as an actual Dps protein (instead of another ferritin-like superfamily protein) can be inferred from the high similarity of the sequences with the functionally active protein previously identified in Saccharolobus solfataricus [83]. In addition, we could not identify any homologous proteins to either Dps or Dps-like as widely present in Euryarchaeota, suggesting that this mechanism could be replaced in the oxidative stress response by another protein of similar function.

#### 3.3.4. Rubrerythrin (Rbr)

Rubrerythrin (Rbr, *rbr*), another member of the ferritin-like superfamily that consists of varied nonheme diiron proteins, plays an important role in oxidative stress response by reducing hydrogen peroxide [85]. Rbr was identified in almost all acidophilic archaea in both aerobic and anaerobic organisms. The identified proteins with rubrerythrin domains are annotated in several different orthogroups, forming distinct clades in the phylogenetic analysis. Five domain architectures were identified across the different proteins. The first group (associated with OG0767) is characterized by a single and isolated Rbr domain with conserved diiron metal motifs; this group was identified across all major phyla. The second group (associated with OG0732) is characterized by a domain architecture of a full Rbr domain with diiron metal motifs together with a VIT1 domain [86]. This group is also conserved across all major phyla under study. A third domain (associated with OG6126, OG7923 and OG3502), similar to the first group identified, contains a small truncated Rbr domain at the C-terminal end and is exclusive to Euryarchaeota. The fourth group (associated with OG1026 and OG3704) presents a truncated Rbr domain with missing diiron metal motifs and with the addition of Linocin M18 domain.

The final domain architecture group (associated with OG1052) presents a small Rbr domain without any diiron metal motifs. The presence of the diiron metal motifs is important in the context of oxidative stress response, as these are essential in the action of Rbr as peroxidase [87]. From these analyses, we propose that groups 1 and 2 may be directly related with hydrogen peroxide reduction. In particular, the addition of the domain VIT1 in group 2 has been previously reported to aid in iron detoxification via transportation of the metals, a process that prevents oxidative stress via Fenton reaction products in plants, yeast and fungi [88,89]. The presence of a Linocin M18 domain in group 3 may aid in the formation of nanocompartments that encapsulate peroxidases [90]. Previously, this variant had only been identified in the extreme thermophile *Pyrococcus furiosus* [85], in which the specific action was hypothesized to be related with the hyperthermophilic trait of these organisms. In our analysis this variant appears not to be exclusive to hyperthermophiles; *Vulcanisaeta* and *Caldivirga* also have this domain architecture and grow at a more moderate temperature (45–50 °C)(Figure 4).

#### 3.3.5. Peroxiredoxins

Peroxiredoxins (Prx) have been identified as the major mechanism of hydrogen peroxide elimination in archaea as reviewed in Pedone et al., 2020. These enzymes have been classified into 6 different subfamilies (AhpC-Prx1, Bcp-PrxQ, Tpx, Prx5, Prx6 and AhpE) using structural information near the active site [91]. A previous study in Saccharolobus solfataricus characterized four different peroxiredoxins from these organisms, three of them (described as Bcp1-3-4) belonging to the Bcp-PrxQ subfamily and one (Bcp2) to the Prx6 [92]. All these sequences have their crystal structure solved, providing experimentally proven and classified examples of peroxiredoxins in archaea [93]. Peroxiredoxins are widely distributed across acidophilic archaea with a varying number of copies (mostly conserved at genus level) depending on the organism. The phylogenetic distribution of the different enzymes identified across the acidophilic organisms is presented in Figure 5a, showing four larger clades that include the proteins characterized in *Saccharolobus* (Prx6/Bcp2, Bcp1, Bcp3 and Bcp4) and a fifth new clade not previously reported in acidophilic archaea with sequences corresponding to the Prx1 subfamily. The clades formed by Bcp1 (PrxQ family) and Bcp2 (Prx6 family) sequences are widely distributed across organisms from all the phyla in study, which shows a high degree of conservation of both families of Prx across all acidophilic archaea. In contrast, Bcp3 is exclusive to organisms of the Crenarchaeota phylum. The Bcp4 clade is divided into three different subclades. The first divergent clade shows a similar distribution to the Bcp3 clade with *Vulcanisaeta*, *Thermoproteus* and *Caldivirga*, the second clade contains sequences from Euryarchaeota (that includes the ARMAN organism *Thermoplasmatales archaeon* B_DKE) and a third clade includes the rest of Crenarchaeota and Marsarchaeota. One last clade, composed of sequences identified as Prx1 and formed by organisms of the phylum Euryarchaeota, Crenarchaeota and Micrarchaeota, represents a divergent branch that is not associated with any of the previously mentioned Saccharolobus sequences. To study this in further detail, multiple sequence alignment of all the sequences in this clade was performed. We identified the presence of the conserved motif PxDFTFVC [90], characteristic of the subfamily Prx1 (Figure 5b). Prx1 from the 2-Cys subfamily acts with a dual enzymatic activity both as peroxiredoxin and catalase [94]. All organisms in which this protein has been identified are either anaerobes or facultative anaerobes, suggesting that they may be utilizing this additional capability of Prx1 in specific conditions and ranges of H_2_O_2_ as has been previously shown for organisms that have different peroxiredoxins [95]. In addition, all organisms with Prx1 have been isolated from similar environments (hot springs or hydrothermal vents), which suggests an econiche adaptation that explains the narrow distribution of Prx1 in comparison with the other Prx conserved across a larger number of acidophilic archaea. The unique divergence of this clade in the phylogenetic tree as well as the identification of the specific motifs confirm the idea that this may be a Prx1 type previously not reported in acidophilic archaea, and further experimental validation is needed to confirm this finding.

#### 3.3.6. Thioredoxin

The mechanism of action of thioredoxins (Trx, *trx*A) and thioredoxin reductases (TR, *trx*B) is similar to that of peroxiredoxins, as they are needed to complete the redox cycle and reduce the disulfide bonds in Prx to continue with the peroxide elimination process. Our results show that most organisms with Trx have two copies, with both copies being part of the same orthogroup (paralogs). This suggests a shared evolutionary story that provides clues of a gene duplication event that may have occurred at an ancestral level. An interesting finding is a hypothetical protein (defined by the orthogroup OG0923) that appears contiguous to *trx*A and was identified in 137 genomes, including members of Crenarchaeota (*Acidianus, Acidilobus*, *Caldivirga*, *Metallosphaera*, *Saccharolobus*, *Sulfolobus*, *Thermocladium* and *Vulcanisaeta*), Euryarchaeota (*Picrophilus* and *Acidiplasma*) and Marsarchaeota with high conservation as shown in representative genomes in Appendix A. No protein domain was identified by functional annotation; all sequences from this orthogroup were analyzed by subcellular localization prediction tools, which predicted that these proteins are in either the cellular membrane or extracellular space, and no signal peptide was identified in the sequences (Appendix A). This highly conserved hypothetical protein may be important for these organisms, as proteins in the cellular membrane of acidophiles are exposed directly to the physicochemical constraints of a low pH environment. Experimental analyses of proteins of unknown function may help us reveal new proteins or mechanisms of oxidative response that are unique to archaea in extreme environments. Predicted TR were also identified widely across acidophilic archaea as part of the same orthogroup, with most members in *Acidianus*, *Saccharolobus*, *Sulfolobus* and Marsarchaeota having an extra copy (Appendix A). The presence of both Trx and TR in addition to different types of Prx suggests that the complete redox thiol cycle may play an important role in oxidative stress response in archaea as has been previously demonstrated in bacteria [96].

### 3.4. Methionine Sulfoxide Reductases

Another target sensitive to ROS oxidation is methionine, a sulfur-containing amino acid. The first stage ROS oxidation is the formation of methionine sulfoxide with two different stereospecific forms (Met-S-O and Met-R-O), a process that may lead to the protein being aggregated and eliminated or even further oxidized to methionine sulfone (an irreversible state). MsrA reduces Met-S-O and MsrB reduces the Met-R-O stereospecific form. The first protein, MsrA, was identified in several genera of the acidophilic archaea. Previous studies have reported this to be mainly absent in hyperthermophiles [97]. In contrast, our results include hyperthermophiles such as *Metallosphaera*, *Acidianus* and *Sulfolobus*, all of which have optimal growth over 70 °C, highlighting a previous underestimation of the abundance of MsrA in archaea. The second protein, MsrB, was only identified in *Thermoplasmatales archaeon* and *Cuniculiplasma* in Euryarchaeota, Marsarchaeota and Thaumarchaeota, completely absent from Crenarchaeota. The protein fRMSR also can reduce the same isoform as MsrB in free amino acids (not as part of a protein) and is characterized by a GAF-like domain. This domain has been identified in *Picrophilus* and *Thermoplasma*, both organisms that also have MsrB. All these proteins share similarity with the fRMSR from *Thermoplasma acidophilum*, where structural and biochemical analysis has been performed to confirm its functional activity [98]. Given the high degree of similarity of the sequences, we suggest that all other copies in the same orthogroup may also be functionally active. Finally, both stereospecific forms can also be reduced by the molybdopterin-dependent sulfite oxidase family MssP. We identified this protein in acidophiles from all phyla. Some organisms from Crenarchaeota (*Saccharolobus*, *Metallosphaera*, *Vulcanisaeta* and *Thermocladium*) have more than one copy, which may act as counterbalance to the fact they lack the other common mechanism for reducing Met-R-O. These results contrast with previous observations that found MsrP to be mostly absent in archaea [97]. MsrQ is commonly used as an electron donor associated with MsrP in bacteria [99]; however, it was not identified in any of the organisms under study, suggesting that an alternative pathway to obtain electrons may be in use by archaea. This may be particularly relevant when considering the differences in cell wall components compared with bacteria. An overview of the described system is shown in Figure 6.

## 4. Discussion

Our study provides valuable evidence about the prediction of the mechanisms used by acidophilic archaea to cope with oxidative stress and represents one of the most extensive comparative genomics analyses on this topic to date. An overview of all the analyzed mechanisms identified is presented in Figure 7. A comparison of the predicted proteins with those from non-acidophilic archaea indicates that most of the mechanisms are conserved widely across the different acidophilic and non-acidophilic phyla in this study. However, there are some interesting deviations from this observation. Superoxide dismutase (SOD) constitutes an interesting case, in which all the structurally described Fe-SOD are from acidophilic archaea, while other non-acidophilic archaea have either Mn-SOD or the cambialistic Mn/Fe-SOD. This difference may be associated with acidophiles maintaining the ancestral version of Fe-SOD [7], given the high availability of iron in acidic environments even in current times, while non-acidophilic archaea prefer SOD that may use other elements as catalytic centers [100]. Peroxiredoxins are also predicted both in acidophilic and non-acidophilic archaea, but one specific family identified in this study (Prx1) has not been previously described in acidophilic archaea in literature, which highlights the importance of the use of comparative genomics strategies to help unravel the distribution of different proteins across a large dataset of organisms. Lastly, a hypothetical protein described by OG0923 was identified to be highly conserved in the genomic context of Trx and restricted to acidophiles. OG0923 is predicted to be transported to the membrane. We hypothesize that it carries out a role in acid stress as membrane proteins are directly confronted by the extremely low pH of the environment. This serves as an example of the use of bioinformatics in predicting potential new proteins associated with the oxidative stress response that may be particular to archaea.

A notable difference that we detect between archaea and bacteria is that both aerobic and anaerobic representatives of acidophilic archaea use SOD to deal with superoxide, whereas anaerobic bacteria use SOR [101]. The archaeal preference for SOD may be related to the issue of the increased availability of Fe in acidophilic environments.

It has been observed that HGT events play an important role in adaptation and speciation in archaea, and that many of these events imply the acquisition of genes from bacteria (mostly acidophilic bacteria) to archaea [102,103]. The high similarity between the sequences from archaea and acidophilic bacteria also highlights that the HGT event could be associated with specific niche adaptations that confer ecological advantage to these organisms. The importance of HGT events in acidophiles has been previously discussed in several works [104,105,106]. Cross-domain events of HGT were also predicted, which could play a role in the evolution of catalases and Dps. In both cases of HGT, we identify a restricted presence in specific lineages of archaea in contrast with what is found in bacteria, where both proteins are widely distributed across the domain [107,108]. This suggests that the most probable case for direction of this HGT event is from bacteria to archaea.

The association of archaea and early anoxic Earth environments and their deep rooting in the tree of life is commonly discussed in literature in the context of archaea as candidates to harbor ancestral mechanisms and the possibility that they may illuminate the origin of different traits [109,110]. The wide conservation across organisms of superoxide dismutase, peroxiredoxin and rubrerythrin in both anaerobic and aerobic organisms may suggest an ancestral origin of these proteins [87,111,112], and that they were evolved in similar environments to the early LUCA in response to local concentrations of oxygen before the GOE [113]. It has been previously reported that multiple mechanisms for oxidative stress response are found in hyperthermophiles [21], and we identify a similar pattern in acidophiles with no major differences across lifestyles (such as aerobic versus anaerobic or thermophiles versus mesophiles). Instead, the distribution of oxidative stress response mechanisms present in each organism seems to be largely determined by phylogeny. Interestingly, the same type of distribution was observed in an extensive study of the stress enzymes in bacteria [11], showing that this could be a trend globally associated with oxidative stress response.

## 5. Conclusions

Through comparative genomics of more than 200 genomes from acidophilic archaea, we identified a wide range of mechanisms they use to cope with oxidative stress mostly conserved across all five different phyla. This comparison showed no clear difference between aerobic and anaerobic lifestyles. In this study, we identify protein families missed in individual studies as the Prx1 family present in several acidophiles. The conservation of two proteins with iron coordinating centers, SOD and Rbr, across almost all organisms in the study suggests an early evolutionary origin that may have arisen in anaerobic early Earth environments and can be maintained in acidic environments. Furthermore, HGT events from bacteria to archaea exemplified in the case of catalases and Dps highlight the importance of these events in prokaryotic adaptation to oxidative stress.

## Figures and Tables

**Figure 1 antioxidants-11-00059-f001:**
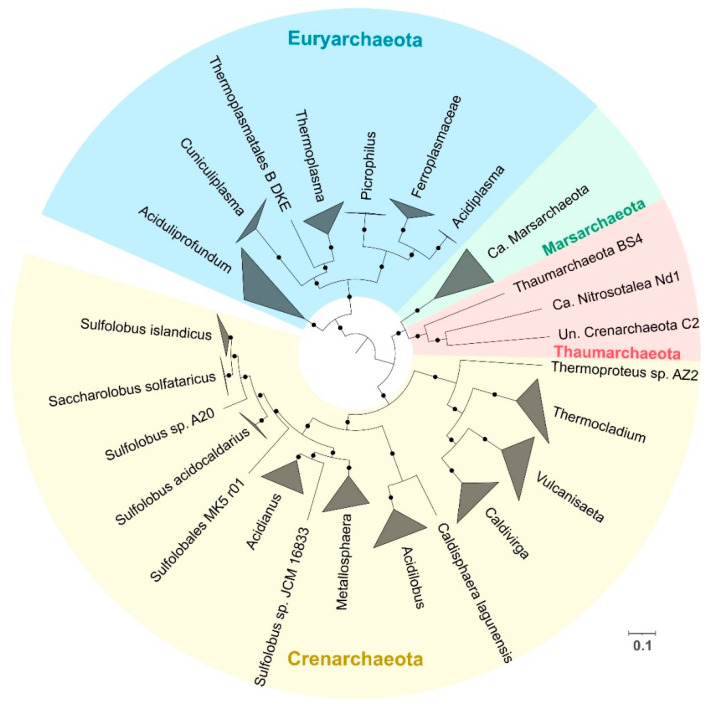
Phylogenetic tree reconstruction of acidophilic archaea with genomes available in AciDB using PhylophlAn 400 markers. Branches are collapsed at genus level, except for *Sulfolobus*, which is collapsed at species level. Colors represent different phyla associations, with yellow for Crenarchaeota, red for Thaumarchaeota, green for Marsarchaeota and blue for Euryarchaeota. Bootstrap values over 60% are represented by black dots. Abbreviations: Uncultured (Un.); Candidatus (Ca.). Scale bar represents 0.1 amino acid substitution per site.

**Figure 2 antioxidants-11-00059-f002:**
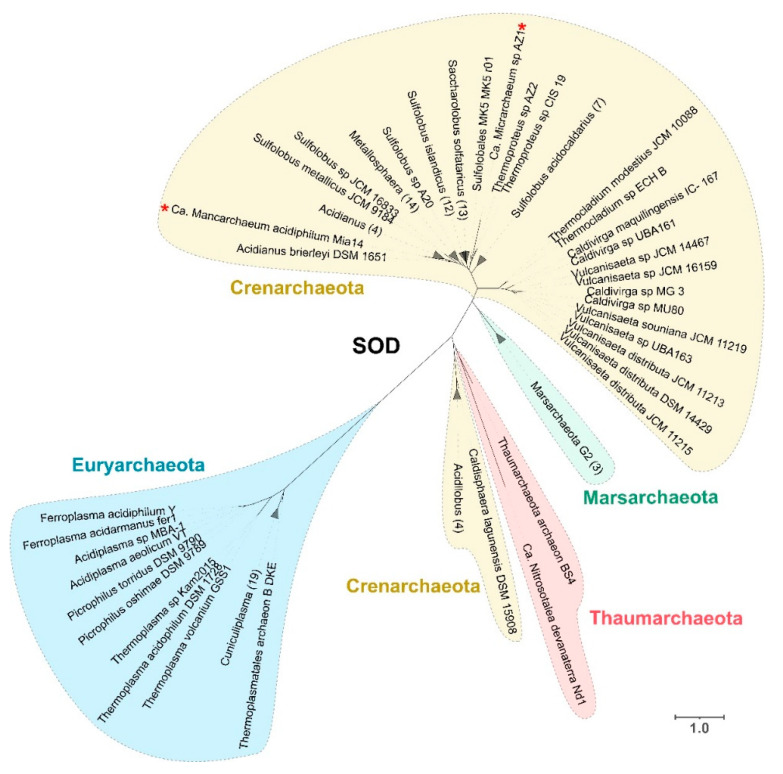
Superoxide dismutase phylogenetic tree from predicted protein sequences for gene *sod*A. Two Candidatus Micrarchaeota leaves are marked with red asterisks, which are in the bigger clade of Crenarchaeota. Colors represent different phyla associations, with yellow for Crenarchaeota, red for Thaumarchaeota, green for Marsarchaeota and blue for Euryarchaeota. Scale bar represents 1 amino acid substitution per site.

**Figure 3 antioxidants-11-00059-f003:**
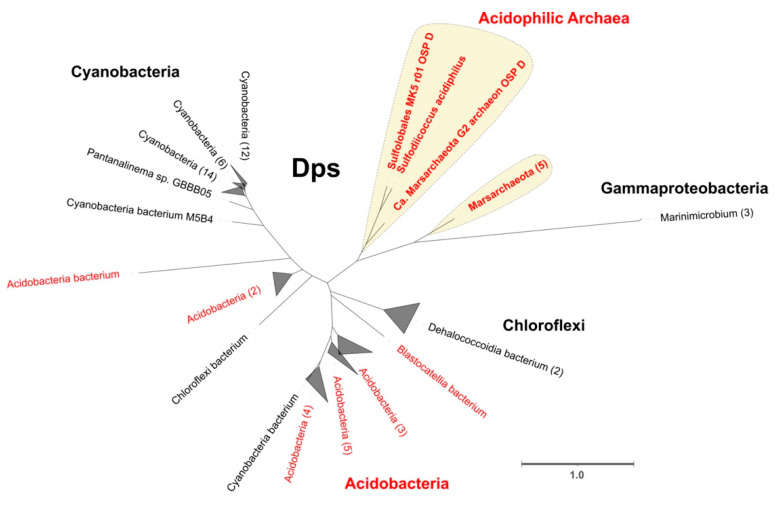
Phylogenetic tree from predicted protein sequences of DNA-binding protein from starved cells (Dps). The protein sequences from acidophilic archaea are indicated with a yellow background. In red letters are the sequences associated with acidophiles. Collapsed branches show the number of genomes in each clade in parenthesis after the names. Scale bar represents 1 amino acid substitution per site.

**Figure 4 antioxidants-11-00059-f004:**
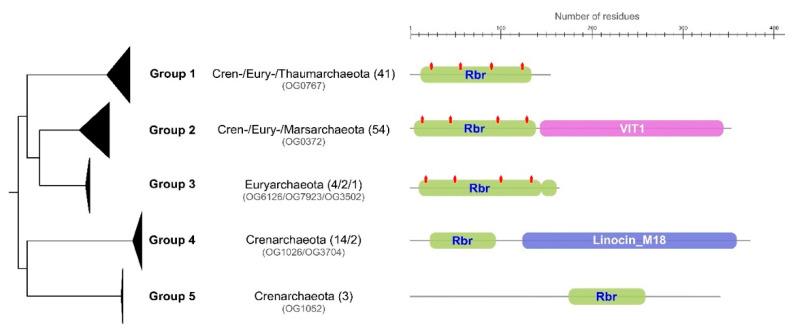
Rubrerythrin domain architecture in acidophilic archaea. The phylogenetic tree is collapsed by clades representing specific protein domain architectures (groups 1–5). Specific orthogroups with phylum association are indicated by each clade with the number of sequences in parentheses. Rbr domains are represented in green, VIT1 domain in purple and Linocin M18 in blue. Diiron metal motifs identified are represented as red markings on Rbr domains. Scale bar represents protein length in residues.

**Figure 5 antioxidants-11-00059-f005:**
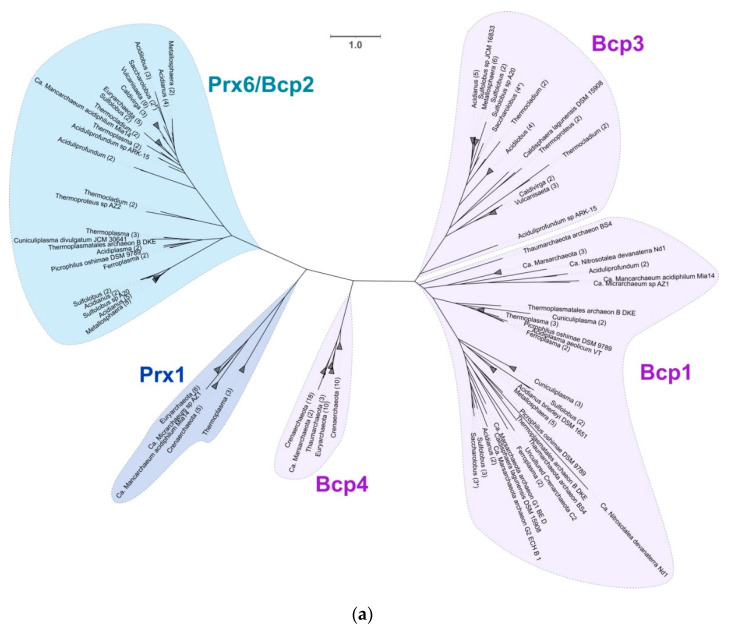
Peroxiredoxin (Prx) family proteins identified across acidophilic archaea. (**a**) Phylogenetic tree of all identified Prx sequences. Branches are collapsed to phyla or genus level when possible (if not, only the names are grouped), with the number of sequences that formed the clade in parentheses (the * indicates *Saccharolobus* with *Sulfolobus* collapsed together). In purple are the sequences from PrxQ type (Bcp1-3-4 like), in blue Prx6 type (Bcp2) and in darker blue Prx1. Scale bar represents 1 amino acid substitution per site. (**b**) Multiple sequence alignment of the Prx1 sequences (N-terminal extract showed). Conserved motifs associated with the characterization of Prx1 are marked in blue and shown in LOGOS graphic in the upper right corner.

**Figure 6 antioxidants-11-00059-f006:**
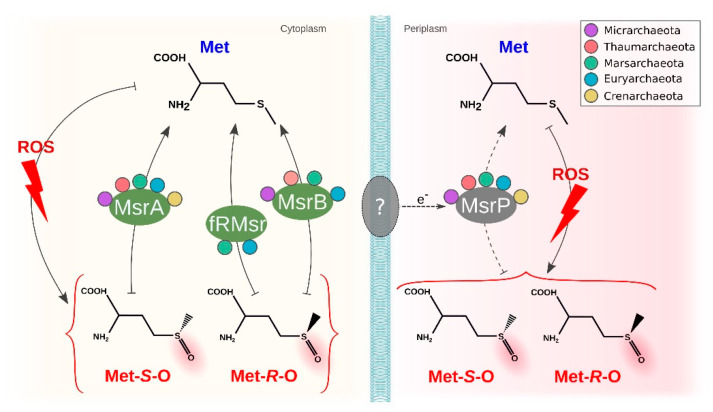
Methionine sulfoxide reductases in acidophilic archaea. Methionine is oxidized by ROS to two different stereospecific forms, where Met-S-O is reduced by MsrA, Met-R-O is reduced by MsrB and fRMsr at the periplasm and both forms are reduced by MsrP obtaining electrons from an unknown source protein. The presence of the proteins in each phylum is represented by circles (purple for Micrarchaeota, pink for Thaumarchaeota, green for Marsarchaeota, blue for Euryarchaeota and yellow for Crenarchaeota).

**Figure 7 antioxidants-11-00059-f007:**
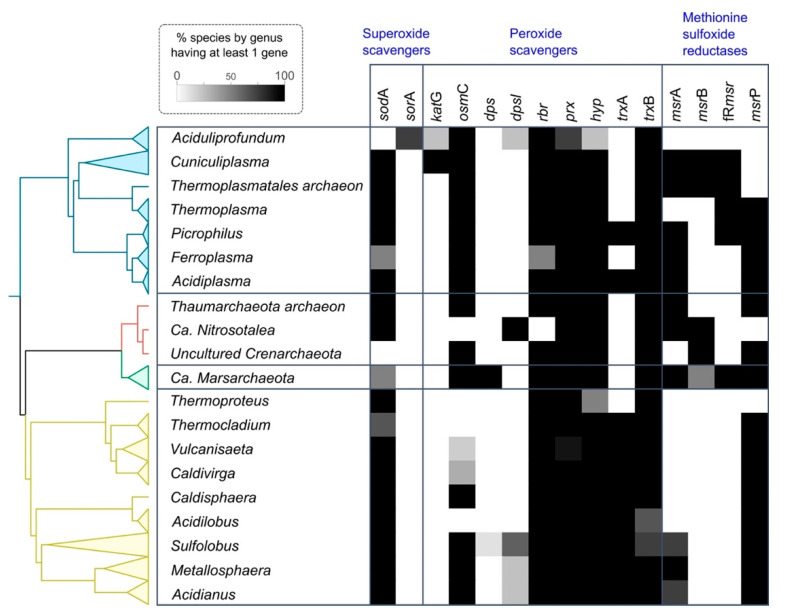
Summary of predicted oxidative stress response proteins in acidophilic archaea. The heatmap depicts the presence of each protein as a percentage of the genomes it was identified in (see insert scale-bar). Genomes were grouped at the genus level. Colors correspond to phylum level with yellow for Crenarchaeota, red for Thaumarchaeota, green for Marsarchaeota and blue for Euryarchaeota.

## Data Availability

The data presented in this study are available in the article.

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
