# Peer review of "A Large-Scale Multiple Genome Comparison of Acidophilic Archaea (pH ≤ 5.0) Extends Our Understanding of Oxidative Stress Responses in Polyextreme Environments"

_antioxidants, 2021, doi:10.3390/antiox11010059_

Round 1

Reviewer 1 Report

This paper describes the oxidative stress responses in Archaea by genome comparison.  This is a potential interesting and important study.  There are, however , concerns that needed to be addressed before publication can be considered. The authors demonstrate horizontal gene transfer has an important role in genome evolution.  The author should show detecting and evaluating of HGT .

Author Response

We have improved the description of the methodology for HGT detection and phylogenetic tree reconstruction in the Material and Methods section by providing references to the techniques used between lines 128-132 and 136-137:

An evolutionary history-based (phylogenetic) approach as described by Ravenhall et al. [56], was used to identify genes whose evolutionary history significantly differs from that of the host species and are inferred to be horizontally transferred genes (HGT). Phylogenetic trees were constructed using the sequences in the orthogroups and their best blastp hits as described by Nelson et al. [57] and Kooning et al. [58]. Proteins from the blastp search were downloaded using the Batch Entrez web tool in NCBI [59]. Sequence alignments were performed using MAFFT and were visualized with Aliview v1.2.6 [60] to identify the conservation of key amino acids when needed. The phylogenetic tree was then constructed using IQTREE with an ultrafast bootstrap of 1000 replicates. Phylogenetic trees were visualized and clades were annotated using iTOL [46]. Genome neighborhoods were analyzed using Gene Graphics with a region size of 5000bp [61].

Reviewer 2 Report

Dear Authors,

Your manuscript ID: antioxidants-1500420 contributes to a better understanding of stress-associated mechanisms in acidophilic Archaea. Manuscript is well written, so I am able to recommended it for publication in Antioxidants journal. I have only two suggestions:

Line 69

Use a capital letter for taxonomic name

Line 447

“acidophilic Archaeas” or “acidophilic Archaea”?

Author Response

We have modified the manuscript according to the suggestions choosing “acidophilic Archaea” in line 470 (before was the line 447).

Reviewer 3 Report

In the manuscript, Gonzalo Neira and co-workers reported a large scale, multiple genomes comparison of acidophilic Archaea (pH ≤ 5.0) extends our understanding of oxidative stress responses in polyextreme environments. Acidophilic Archaea detect in different anaerobic and aerobic low pH environmental ecosystems, rich in dissolved heavy metals which may exacerbate stress caused by the production of reactive oxygen species (ROS)  such as hydrogen peroxide, hydroxyl radical and superoxide. These organisms can be subjected to simultaneous multiple stresses such as high temperature, high salinity, low pH and high heavy metal loads. In manuscript present big massive investigation data processed using modern bioinformatic methods. Over 200 genomes of acidophilic Archaea were used for phylogenomic distribution genes potentially involved in mitigation (ROS). Authors examine in manuscript following topics: (1) the phylogenomic distribution of these genes and what can this tell us about the evolution of these mechanisms in acidophilic Archaea; (2) key differences in genes and mechanisms used by acidophilic versus non-acidophilic Archaea and between acidophilic Archaea and acidophilic Bacteria and (3) how comparative genomic analysis predicts novel genes or pathways involved in oxidative stress responses in Archaea and likely Horizontal Gene Transfer (HGT) events. As a comment in the following articles use the new classification of microorganisms according to Rinke et al., 2020 (A rank-normalized archaeal taxonomy based on genome phylogeny resolves widespread inconplete and uneven classifications).

Author Response

We thank the reviewer for the suggestion and plan to use the rank-normalized taxonomy in subsequent research.

Reviewer 4 Report

The manuscript written by Neira C et al describes comprehensive analysis of the genes related to oxidative stress responses in the genomes of acidophilic Archaea and proposed that their analyzed data could provide useful insight into the evolution of oxidative stress responses in exposure to oxygen as the environment shifted from anaerobic to aerobic. This manuscript just includes bioinformatic analysis data and any experimental result is shown. However, I agree with that the comprehensive search for the related genes on the genomes of the sequenced acidophilic archaea and comparison of their distributions with those of other archaea and bacteria will be useful to think about evolution and functions. I have just a few comments, which I hope to improve the manuscript.

  1. Their analysis showed that all the acidophilic archaea had Fe-SOD, and the discussion that the higher availability of iron in low pH environments is interesting. I think it will be more interesting to add some description of the distribution of other type of SOD and how distant between the different types of SOD evolutionarily.

  1. Analysis of SOR is shown by the sequence comparison of SorA from sorGOdb. The names of SorA snd sorGOdb should be explained clearly for the readers not familiar with this field.

  1. A new finding of this work is that a conserved gene, OG0923, is found in the position next to Furthermore, the encoded protein is predicted as a membrane protein that may be important for acidophilic phenotype of these organisms. I think it is an interesting finding and recommend the authors to move Fig S5 from suppl to the main text, and more descriptions of the OG0923 protein including more precise distribution of the gene and more detailed features of the protein predicted from the amino acid sequence.  

Author Response

1.We have now adressed the distribution of SOD in non-acidophilic Archaea and have added references 69,70 to strengthen the concept (lines 187 - 189).  

  1. The complete name of the database was added (Lines 224-225).With regard to the use of SorA, we decided to change Table S2 reference of sorA to the general denomination of sor in order to avoid any possible confusion. The classification results in the comparison with SORGOdb is now reflected in line 226.
  2. Additional details of the distribution of OG0923 and its description was added in lines 416-420 and lines 422-423, respectively. We would like to retain Figure S5 in the supplementary files as we feel that it is ancilliary but yet important to the main story of the paper

Reviewer 5 Report

The paper is in general well written and presented and includes only minor language errors, such as missing articles etc. in few circumstances (e.g. line 177-" in this study").

The figures resolution could be of better quality to help the reader and maybe font size of the figures could be increase where relevant (e.g. Fig. 5).

There are no further ocmments or improvements on the scientific content.

Author Response

Cases like “in study” of line 183 and in lines 326, 328 are reference to data presented in our article and therefore no article was referenced. The previous mentioned lines were changed to provide more clarity that we are referring to data of our study (using e.g. “under study”)

Phylogenetic branches and their names have been collapsed (e.g. Fig 5) without compromising the data of the study. Branch collapse makes the Figs more readable. Font size was increased from 10 to 12. Resolution was improved from 300dpi to 500dpi giving a clearer image if zoomed.